# Anemia-driven heterogeneity in TG-UA Association: A DDML approach to precision management of hyperuricemia in hospitalized patients aged 50–65

Tong Zhi[1,2], Jin Song[3], Shuai Zhang[1], Qi Li[1], Mingrui Li[1], Huaxin Zhang🔾[1]*

**1** Beijing Fengtai Youanmen Hospital, Beijing China, **2** School of Government, Beijing Normal University, Beijing, China, **3** Beijing GoBroad Boren Hospital, Beijing, China

* 18831345670@139.com

## Abstract

Hyperuricemia has posed a great threat in the globe and is also linked with serious health risks such as hyperlipidemia. Extensive researches have explored the relationship between hypertriglyceridemia and serum uric acid (UA), but the results are mixed due to multiple demographic and cardiovascular-related factors. Our research aims to delve into the relationship between triglyceride (TG) and serum UA across various demographic groups and groups with different anemic conditions. Our research concentrated on patients admitted to Beijing Fengtai Youanmen Hospital from August 2016 to June 2023, the final dataset encompassed clinical data from 43,758 patients, providing a robust basis for our analysis. Data analysis was conducted using Stata MP version 17.1 (Stata Corp LLC, College Station, TX, USA). By using double debiased machine learning method to delve into the relationship between TG and UA. Our findings indicate a strong association between elevated TG levels and increased UA levels among individuals aged 50–65 years ($P < 0.001$). Our results showed a positive relationship between TG and UA in non-anemic males aged 50–65 when TG levels were 1.1–1.90 mmol/L, and similarly among non-anemic females aged 50–65, TG-UA correlation persisted with TG levels below 2.30 mmol/L. Conversely, among anemic patients aged 50–65, both in male and female groups, there is no significant association between TG and UA levels. Our research suggests that to address hyperuricemia, reducing blood TG levels may be an effective strategy. However, this strategy should not be applied to anemic patients aged 50–65.

## Introduction

Hyperuricemia, acknowledged globally as a key metabolic disease, is linked with serious health risks such as hypertension, hyperglycemia, and hyperlipidemia [1]. It is diagnosed by specific serum UA levels—above 417 µmol/L among male group and 357 µmol/L among female group—and is associated with disorders like gout, cardiovascular diseases, and diabetes [2–6]. The prevalence and burden of hyperuricemia are increasing, particularly in affluent regions and some developing countries [7,8]. The researches focusing on hyperuricemia is

**Data availability statement:** Data cannot be shared publicly because of the data are owned by Beijing Fengtai Youanmen Hospital and authors do not have permission to share the data. However, researchers who meet the criteria for accessing confidential data can request access from the Youanmen Hospital Ethics Committee (contact: yamyyirb@126.com or Tel: +86-010-83322178). This study was approved by the Beijing Fengtai Youanmen Hospital Ethics Committee Institutional Review Board (Approval No. LL-2024-23).

**Funding:** The author(s) received no specific funding for this work.

**Competing interests:** The authors have declared that no competing interests exist.

increasing due to its growing prevalence and the significant health and economic impacts it imposes globally [9,10]. Studies have shown the prevalence of hyperuricemia found is higher in most developed countries, the prevalence of hyperuricemia is higher in coastal areas and countries than in landlocked countries, especially for countries surrounded by sea and in developing [11]. Survey (NHANES) 2007–2016, a nationally representative survey showed that the prevalence rates of hyperuricemia were 20.2% among men and 20.0% among women between 2015–2016 in the United States and the incidence of hyperuricemia remained stable in 2007–2016 [9]. And by 2014, the overall prevalence of hyperuricemia in mainland China had reached 13.3%, and this gradually increased to 17.7% in 2017 [12,13]. Substantial evidence suggests that hyperuricemia is mechanistically associated with dyslipidemia, particularly through shared pathways involving insulin resistance, oxidative stress, and lipid metabolism dysregulation [14]. This bidirectional relationship implies that targeted management of dyslipidemia—via lifestyle interventions or lipid-lowering therapies—could synergistically mitigate hyperuricemia progression, thereby reducing associated cardiovascular and renal risks [15–19].

Researches exploring the relationship between TG and UA have been extensive, but the results are mixed. On one hand, Tan et al. [2023], employed logistic regression and other statistical models to analyze the data from the National Health and Nutrition Examination Survey in China, confirming a strong association between elevated serum UA levels and hypertriglyceridemia [20]. Similarly, Setiawan and Lestari [2023] identified a significant positive correlation between these two biomarkers in patients with coronary heart disease [21]. On the other hand, in some other studies, no association between SUA and TGs were found in multivariable analyses [22]. The potential reasons for the differences in results may lie in the influence of fundamental factors such as gender, age [23], and metabolic-related factors [24] on the relationship between TG and UA. These factors need to be fully explored in relation to this association.

In this paper, we propose a method called Double/Debiased Machine Learning (DDML). This approach offers several advantages. First, machine learning models generally outperform traditional statistical methods, such as difference-in-differences, in terms of predictive accuracy. Second, machine learning frameworks relax the rigid assumptions inherent in traditional models, enabling the direct learning of relationships between variables from the data [25]. Grounded in machine learning algorithms, DDML is distinguished by its use as an unbiased estimator, which is crucial for its credibility in data analysis. Notably, DDML is capable of capturing not only causal linear relationships but also fully non-linear effects [26], highlighting its versatility and robustness in addressing complex analytical challenges.

We adoption of DDML as the primary analytical tool is underpinned by its capacity to integrate causality within machine learning frameworks. This integration not only enables the direct extraction of relationships between input and output variables from datasets but also facilitates the effective mitigation of confounding effects, thereby bolstering the authenticity of the cause-effect relationships identified [27,28]. A prominent attribute of these models is their ability to maintain interpretability, empowering modelers to thoroughly examine the impacts of various causal factors or policies. This advancement represents a significant leap in computational data analysis, effectively bridging the divide between empirical data and causal inference.

Therefore, our research aims to delve into the relationship between TG and UA across various demographic groups by DDML method. Moreover, the specific role of lipids under certain thresholds has not been thoroughly investigated. Our research seeks to identify if there is a particular TG threshold that results in elevated UA levels in different populations.

According to the results, specific recommendations for the treatment and daily management of hyperuricemia based on multiple TG thresholds has been provided.

## Materials and methods

### Study sample

Our study is a retrospective analysis, where we obtained patient data from Beijing Fengtai Youanmen Hospital spanning from August 1, 2016 to June 30, 2023 on March 5, 2024. As a comprehensive tertiary medical center within a medical consortium that includes several hospitals in Beijing, this institution facilitates bidirectional referral services with numerous top-tier hospitals. The patient cohort was diverse, encapsulating a broad spectrum of medical conditions. The data selection process was meticulously structured and involved multiple stages:

Firstly, exclusion of records with incomplete information on outcome or independent variables. Removal of records lacking essential demographic details such as gender or age.

Secondly, data features, including comorbidity history or the number of previous laboratory tests, were typically complete (accounting for "not present" or "zero" as valid values). For missing numerical data, we imputed values using the training sample mean. To address the influence of outliers, a 1% winsorization was applied to crucial continuous variables.

Thirdly, in instances of multiple entries for a single patient, only the initial record was incorporated into the dataset. Following these stringent criteria, the final dataset encompassed clinical data from 43,758 patients, providing a robust basis for our analysis.

Our research adheres to the ethical principles outlined in the Declaration of Helsinki. The ethical committees have granted their approval for the research protocol. Furthermore, this trial is officially registered at Beijing Fengtai Youanmen Hospital under the identifier LL-2024-17.

### Measures

In our research, the primary outcome variable was UA, while serum TG were designated as the primary independent variable. To scrutinize its impact, serum TG levels in the dataset were meticulously organized in ascending order. Following this, these values were stratified into distinct categories according to a set of predefined percentiles: 10%, 20%, 30%, 40%, 50%, 60%, 70%, 80%, and 90%. Subsequently, each TG level was dichotomized: values surpassing the respective percentile thresholds were labeled as "1", signifying elevated TG levels, whereas those falling below were marked as "0". This approach is pivotal as it facilitates a detailed examination of serum TG's effect, accommodating the variable concentrations within the patient cohort under research.

Our research also controls for a series of covariates that might have an impact on UA metabolism: age, gender, glucose (GLU), creatinine (CREA), cholinesterase (CHE), creatine kinase-MB mass (CKMB), hemoglobin (HGB). The estimated events were adjusted with up to 7 covariates.

### Statistical analysis

Data analysis was conducted using Stata MP version 17.1 (Stata Corp LLC, College Station, TX, USA), with two-sided p-values less than 0.05 deemed statistically significant.

DDML represents a sophisticated framework engineered to enhance the precision of estimates in predictive models, particularly beneficial in scenarios involving high-dimensional data. As detailed in existing literature [29], DDML aims to rectify the bias introduced through

overfitting when assessing causal effects or constructing predictive models in environments characterized by high-dimensional data. The application of DDML extends across various domains, notably in estimating the effects of health interventions and the impact of policies. For instance, one research utilized DDML to ascertain weighted cumulative treatment effects in time-to-event outcomes, showcasing its applicability in observational studies where randomized controlled trials may not be viable [30]. Additionally, DDML's usage in evaluating mobile health (mHealth) intervention policies further underscores its value in the realms of personalized medicine and health policy assessment [31].

The process of DDML can be divided into two key stages [29]: 1) Nuisance Parameter Estimation, this stage focuses on estimating confounding variables that are not of primary interest but may influence the outcome. The corresponding formula is as follows: $\breve{\eta}\ (X) =$ Machine Learning Estimation $(X)$. Here, $X$ represents control variables. 2) Primary Parameter Estimation, this stage is concerned with estimating the effect of the treatment variable, adjusted for the nuisance parameters. The stages can be expressed by the following formula: $\breve{\theta} = \mathrm{E}[Y - \breve{\eta}(X)|D]$, where $Y$ is the outcome variable, and $D$ is the treatment variable.

In our research, we employed the DDML method to delve into the relationship between TG and UA, recognizing that UA levels are influenced by a range of covariates, including liver function, renal function, cardiac function, HGB, and glucose levels. DDML adeptly addresses the issue of canonical bias through the secondary application of machine learning models in auxiliary equations. This approach effectively mitigates the challenges associated with the high dimensionality that traditional linear regression encounters due to an overabundance of control variables. Moreover, DDML enhances the precision of our estimations by employing a nonparametric machine learning model to navigate nonlinear relationships. This method does not require the predetermination of the relationship between variables, allowing for a more nuanced and accurate exploration of the data [29].

## Results

### Descriptive analysis

Our research encompassed a total of 43,758 participants, comprising 25,229 men and 18,529 women. The analysis revealed no significant disparity in the average age between the male and female cohorts. Laboratory test outcomes indicated distinct variations in UA levels between male and female inpatients, with a statistical significance ($P < 0.001$). Analogous disparities were noted in TG, glucose (GLU), creatinine (CREA), and cholinesterase (CHE) levels. However, sodium ion analysis demonstrated no substantial difference in electrolyte levels between the genders. Additionally, our dataset shows a significant difference in hemoglobin (HGB) levels among individuals aged 50–65 years ($P < 0.001$), whereas creatine kinase-MB mass (CKMB) levels exhibited no variation within this age bracket. (Table 1)

### DDML

In our research, we have developed a partially-linear deep DDML model, where the impact of TG on UA is treated as linear. This model leverages machine learning algorithms to capture the influence of confounding variables through non-parametric and flexible functions. Additionally, we utilize neural network estimators to model the effects of other variables that influence both TG and UA. The optimal hyperparameters for these neural network estimators were determined using 5-fold cross-validation, with the results detailed in Table 2.

We adjusted for interactions of gender, age, GLU, CREA, CHE, CKMB-mass, HGB, Sodium ion, and all variables. TG were identified as a significant risk predictor for UA levels, particularly when categorized at or above the 30th percentile. It was observed that TG levels

**Table 1. Comparison of population characteristics and clinical data.**

| Variable | | Male N=25,229 | | Mean | Female N=18,529 | | Mean | t |
|---|---|---|---|---|---|---|---|---|
| | | n | Ratio | | n | Ratio | | |
| Age<=50 | | 10,243 | 62.52% | 36 | 6,140 | 37.48% | 36 | χ2= |
| 50<age<=65 | | 8,160 | 58.72% | 58 | 5,737 | 41.28% | 58 | 436.63 |
| Age>65 | | 6,826 | 50.65% | 73 | 6,652 | 49.35% | 76 | *** |
| UA | | 25229 | 57.66% | 385.23 | 18529 | 42.34% | 316.98 | 55.99 |
| | | | | | | | | *** |
| TG | | 25229 | 57.66% | 1.76 | 18529 | 42.34% | 1.53 | 16.86 |
| | | | | | | | | *** |
| GLU | | 24955 | 57.69% | 8.13 | 18301 | 42.31% | 7.92 | 5.12 |
| | | | | | | | | *** |
| Sodium ion | | 23766 | 59.43% | 141.06 | 16226 | 40.57% | 141.03 | 0.57 |
| CREA | | 25229 | 57.66% | 93.91 | 18528 | 42.34% | 72.92 | 21.31 |
| | | | | | | | | *** |
| CHE | | 23767 | 59.41% | 7701.89 | 16240 | 40.59% | 7603.14 | 4.36 |
| | | | | | | | | *** |
| 50<age<=65 | CKMB ng/ml | <=2 | 1732 | 58.53% | 2.00 | 1227 | 41.47% | 2.00 | / |
| | | >2 | 1688 | 71.89% | 11.34 | 660 | 28.11% | 10.35 | 0.67 |
| | HGB g/L | <=115 | 611 | 51.60% | 99.23 | 573 | 48.40% | 102.65 | -4.46*** |
| | | >115 | 7190 | 59.61% | 148.67 | 4872 | 40.39% | 136.13 | 51.63*** |

a Abbreviations: creatinine (CREA), creatine kinase-MB mass (CKMB mass), cholinesterase (CHE), glucose (GLU), hemoglobin (HGB), triglycerides (TG), uric acid (UA).

b Significance levels are denoted as follows: *** $P < 0.001$.

**Table 2. Association of TG levels with odds of elevated UA.**

| Explanatory Variable | TG mmol/L | Coef. | 95% CI | |
|---|---|---|---|---|
| 10th percentile of TG | 0.600 | -0.413 | -2.558 | 1.731 |
| 20th percentile of TG | 0.800 | -4.291 | -8.616 | 0.034 |
| 30th percentile of TG | 1.000 | 15.902 | 11.541 | 20.263 |
| 40th percentile of TG | 1.100 | 10.765 | 5.855 | 15.675 |
| 50th percentile of TG | 1.300 | 2.106 | -0.252 | 4.464 |
| 60th percentile of TG | 1.500 | 21.835 | 13.886 | 29.784 |
| 70th percentile of TG | 1.800 | 19.755 | 2.563 | 15.880 |
| 80th percentile of TG | 2.200 | 22.141 | 13.387 | 30.896 |
| 90th percentile of TG | 2.900 | 8.923 | -2.667 | 20.514 |

a. Horizontal lines are 95% CI. Gender, Age (continuous variable), GLU, CREA, CHE, CKMB mass, HGB are controlled in each model

b. Description of each variable in the formula: Y (outcome variable: UA), D (treatment variable: TG level), X (control variables).

exceeding 1.00 mmol/L (95% CI:11.54–20.26) contribute to an increase in UA, a trend that begins to diminish and becomes less significant at approximately 2.90 mmol/L (95% CI: -2.67–20.51) (Table 2).

The research also revealed a notable gender difference in the impact of TG on UA levels. Specifically, in the male cohort, TG exhibited a positive influence on UA for levels exceeding 0.70 mmol/L (95% CI:16.99–10.55). Conversely, in the female group, a positive effect of TG on UA was only observed when TG levels were above 1.10 mmol/L (95% CI: 13.07–31.83). This distinction underscores a significant sex-based variation in the relationship between TG and UA. (Table 3).

**Table 3. Gender differences in the Association of TG levels with odds of elevated UA.**

| Explanatory Variable | TG mmol/L | Male Coef. | 95% CI | | TG mmol/L | Female Coef. | 95% CI | |
|---|---|---|---|---|---|---|---|---|
| 10th percentile of TG | 0.700 | 16.986 | 10.545 | 23.427 | 0.600 | 3.351 | -6.034 | 12.735 |
| 20th percentile of TG | 0.800 | 18.030 | 10.271 | 25.790 | 0.800 | 29.155 | 20.759 | 37.552 |
| 30th percentile of TG | 1.000 | 23.176 | 16.519 | 29.834 | 0.900 | -1.683 | -6.544 | 3.179 |
| 40th percentile of TG | 1.100 | 11.939 | 0.974 | 22.905 | 1.100 | 22.454 | 13.074 | 31.835 |
| 50th percentile of TG | 1.300 | 12.543 | 6.127 | 18.959 | 1.200 | 18.227 | 12.284 | 24.171 |
| 60th percentile of TG | 1.500 | 17.082 | 5.649 | 28.516 | 1.400 | 16.921 | 11.104 | 22.738 |
| 70th percentile of TG | 1.800 | 18.430 | 8.941 | 27.919 | 1.700 | 16.053 | 10.275 | 21.832 |
| 80th percentile of TG | 2.300 | 15.672 | 6.212 | 25.131 | 2.000 | 22.874 | 15.315 | 30.433 |
| 90th percentile of TG | 3.200 | 12.500 | 5.622 | 19.378 | 2.600 | -9.542 | -14.174 | -4.911 |

a. Horizontal lines are 95% CI. Age (continuous variable), GLU, CREA, CHE, CKMB mass, HGB are controlled in each model.

b. Description of each variable in the formula: Y (outcome variable: UA), D (treatment variable: TG level), X (control variables).

Our research yielded a significant observation within the patient aged 50–65 years. Within this age bracket, elevated TG levels were consistently correlated with increased UA levels, a trend that persisted across both genders (Table 4).

Furthermore, our findings indicate no correlation between TG and UA in anemic patients aged 50–65 years. However, in non-anemic male patients within the same age group, a positive relationship between TG and UA was identified when TG levels ranged from 1.1 to 1.90 mmol/L. In non-anemic female patients aged 50–65 years, TG and UA were positively correlated as long as the TG levels were below 2.30 mmol/L (Table 5).

## Discussion

Our research systematically investigated the impact of TG on UA across a diverse population differentiated by gender and age. Utilizing descriptive analysis, we noted variances in UA levels between male and female inpatients. DDML statistical method revealed that in the general population, an increase in TG levels beyond a specific threshold was significantly associated with worsened UA levels. Our research highlighted notable findings within the demographic aged 50–65 years. In this age group, elevated TG levels were consistently linked to increased UA levels, a trend observed in both genders, as detailed in Table 4. Additionally, our results showed that in non-anemic male patients within this age bracket, a positive relationship between TG and UA was evident when TG levels ranged from 1.1 to 1.90 mmol/L. In non-anemic female patients aged 50–65 years, a positive correlation between TG and UA persisted as long as TG levels below 2.30 mmol/L, as illustrated in Table 5. Among anemic patients aged 50–65, both in male and female groups, there is no significant association between TG and UA levels.

Our findings, which reveal significant gender differences in the impact of TG on UA levels, are consistent with multiple studies. For instance, research in a Korean cohort demonstrated that elevated serum UA levels were linked to increased total cholesterol and TG in women, but not in men [32]. Likewise, Devi et al. [2020] found that elevated serum UA levels were more prevalent among females compared to males within the context of metabolic syndrome [33]. Zheng et al. [2021] further identified gender-specific effects, noting that UA significantly mediated the relationship between waist circumference and the TG glucose index more in women than in men [34]. These findings suggest that females might exhibit a more sensitive biochemical response to fluctuations in TG levels, possibly influenced by hormonal differences, variations in fat distribution, and metabolic factors that uniquely affect women. Such

**Table 4. Gender and age differences in the association of TG with elevated UA.**

| Explanatory Variable | Male | | | | | | | | | Female | | | | | | | | |
|---|---|---|---|---|---|---|---|---|---|---|---|---|---|---|---|---|---|---|
| | TG mmol/L | Age<=50 | | TG mmol/L | 50<age<=65 | | TG mmol/L | Age>65 | | TG mmol/L | Age<=50 | | TG mmol/L | 50<age<=65 | | TG mmol/L | Age>65 | |
| | | Coef. | P | | Coef. | P | | Coef. | P | | Coef. | P | | Coef. | P | | Coef. | P |
| 10th percentile of TG | 0.700 | 14.080 | 0.126 | 0.700 | 3.591 | 0.000 | 0.600 | 5.380 | 0.396 | 0.500 | -20.974 | 0.459 | 0.800 | 9.420 | 0.019 | 0.700 | 34.612 | 0.000 |
| 20th percentile of TG | 0.800 | 5.790 | 0.230 | 0.900 | 3.590 | 0.000 | 0.700 | -4.302 | 0.524 | 0.700 | -8.639 | 0.216 | 0.900 | 13.853 | 0.000 | 0.900 | -0.450 | 0.903 |
| 30th percentile of TG | 1.000 | 6.370 | 0.243 | 1.000 | 4.209 | 0.000 | 0.900 | 11.538 | 0.002 | 0.800 | -4.592 | 0.474 | 1.100 | 10.891 | 0.001 | 1.000 | 19.999 | 0.000 |
| 40th percentile of TG | 1.200 | 4.754 | 0.248 | 1.200 | 4.305 | 0.000 | 1.000 | 1.382 | 0.657 | 0.900 | 1.653 | 0.800 | 1.200 | 12.614 | 0.000 | 1.100 | 12.202 | 0.004 |
| 50th percentile of TG | 1.400 | 14.320 | 0.067 | 1.400 | 3.902 | 0.000 | 1.100 | 7.992 | 0.036 | 1.000 | -3.020 | 0.713 | 1.400 | 12.499 | 0.000 | 1.300 | 24.943 | 0.000 |
| 60th percentile of TG | 1.700 | 18.310 | 0.011 | 1.600 | 4.354 | 0.000 | 1.300 | 3.541 | 0.146 | 1.200 | 3.794 | 0.771 | 1.600 | 17.986 | 0.000 | 1.500 | -0.585 | 0.861 |
| 70th percentile of TG | 2.100 | 9.145 | 0.141 | 1.900 | 4.265 | 0.000 | 1.500 | 2.584 | 0.402 | 1.400 | 6.336 | 0.696 | 1.900 | 26.849 | 0.000 | 1.700 | 27.400 | 0.000 |
| 80th percentile of TG | 2.700 | 3.716 | 0.360 | 2.300 | 3.572 | 0.000 | 1.700 | 2.482 | 0.339 | 1.700 | 3.378 | 0.840 | 2.200 | 10.044 | 0.045 | 2.000 | -3.096 | 0.344 |
| 90th percentile of TG | 3.800 | -0.742 | 0.817 | 3.200 | 4.188 | 0.000 | 2.300 | -9.645 | 0.002 | 2.300 | 11.286 | 0.527 | 2.900 | 12.577 | 0.001 | 2.600 | 0.501 | 0.928 |

a. Horizontal lines are 95% CI. GLU, CREA, CHE, CKMB mass, HGB are controlled in each model.

b. Description of each variable in the formula: Y (outcome variable: UA), D (treatment variable: TG level), X (control variables).

**Table 5. Relationship between TG and UA 50-65 group (with gender and HGB difference).**

| Explanatory Variable | Male 50<age<=65 | | | | | | Female 50<age<=65 | | | | | |
|---|---|---|---|---|---|---|---|---|---|---|---|---|
| | TG mmol/L | HGB<=115g/L | | TG mmol/L | HGB>115g/L | | TG mmol/L | HGB<=115g/L | | TG mmol/L | HGB>115g/L | |
| | | Coef. | P | | Coef. | P | | Coef. | P | | Coef. | P |
| 10th percentile of TG | 0.700 | -0.187 | 0.619 | 0.700 | -2.755 | 0.499 | 0.700 | 0.062 | 0.865 | 0.800 | 6.395 | 0.003 |
| 20th percentile of TG | 0.800 | -0.193 | 0.608 | 0.900 | 6.183 | 0.224 | 0.800 | 0.080 | 0.826 | 1.000 | 7.392 | 0.000 |
| 30th percentile of TG | 1.000 | -0.184 | 0.624 | 1.100 | 14.316 | 0.005 | 1.000 | 0.073 | 0.841 | 1.100 | 7.841 | 0.000 |
| 40th percentile of TG | 1.100 | -0.182 | 0.627 | 1.200 | 11.070 | 0.001 | 1.100 | 0.076 | 0.833 | 1.300 | 8.242 | 0.000 |
| 50th percentile of TG | 1.300 | -0.178 | 0.637 | 1.400 | 18.408 | 0.002 | 1.300 | 0.076 | 0.833 | 1.400 | 8.393 | 0.000 |
| 60th percentile of TG | 1.500 | -0.181 | 0.632 | 1.600 | 6.898 | 0.124 | 1.500 | 0.068 | 0.850 | 1.600 | 4.280 | 0.015 |
| 70th percentile of TG | 1.700 | -0.178 | 0.637 | 1.900 | 11.926 | 0.030 | 1.800 | 0.070 | 0.847 | 1.900 | 7.139 | 0.000 |
| 80th percentile of TG | 2.000 | -0.190 | 0.614 | 2.300 | 0.345 | 0.935 | 2.200 | 0.088 | 0.807 | 2.300 | 5.283 | 0.026 |
| 90th percentile of TG | 2.700 | -0.196 | 0.602 | 3.300 | 8.061 | 0.258 | 3.000 | 0.077 | 0.831 | 2.900 | 3.831 | 0.148 |

a. Horizontal lines are 95% CI. GLU, CREA, CHE, CKMB mass are controlled in each model.

b. Description of each variable in the formula: Y (outcome variable: UA), D (treatment variable: TG level), X (control variables).

mechanisms could involve the role of estrogen in lipid metabolism and insulin sensitivity, potentially intensifying the correlation between TG and UA more pronouncedly in females than in males.

Our research findings underscore a consistent association between elevated TG and increased UA levels across both genders within the 50-to-65-year population, echoing a significant body of existing literature. Numerous studies, such as those by Ma et al. [2019] and Zhao et al. [2005], consistently report a positive TG-UA association across various age groups, including individuals aged 50–65, suggesting this is a widespread physiological phenomenon [35,36]. This association is also evident in specific patient populations, such as those with metabolic syndrome [33] and clear cell renal cell carcinoma (ccRCC) [37], indicating that the TG-UA correlation extends across different health conditions. A plausible underlying mechanism can be that UA results from purine metabolism. Specifically, increased TG can enhance purine turnover [38] or reduce UA's renal excretion [39], leading to hyperuricemia. Another plausible explanation for the result can be both TG and UA contribute to insulin resistance. High insulin can reduce UA excretion and increase TG levels. Besides, chronic inflammation linked with high TG can also affect UA metabolism. Despite our result consists with prior researches, variations may occur due to population-specific factors (e.g., differences in diet, genetics, lifestyle and etc.), comorbid conditions (e.g., renal dysfunction, cardiovascular diseases) and methodological differences (e.g., study design, sample size, and measurement techniques).

Anemia may influence factors that modulate the relationship between TG and UA levels among patients aged 50–65 in our study. Although anemia disrupts various metabolic functions, including oxygen transport, kidney function, and hormonal levels, these effects do not appear to result in a significant correlation between TG and UA levels in this population. Chronic anemia often leads to hypoxia, shifting cellular metabolism from aerobic to anaerobic pathways. This metabolic shift results in increased UA production due to the breakdown of ATP, which could also alter TG levels [40,41]. Additionally, anemia related to renal dysfunction can alter lipid metabolism by promoting lipid accumulation and aggravating conditions such as atherosclerosis, particularly in hemodialysis patients [42,43]. Renal anemia, with impaired kidney function, can also reduce UA excretion, leading to hyperuricemia—a common issue in these patients that can exacerbate conditions like gout [44]. Furthermore,

hormonal changes associated with anemia, such as variations in testosterone levels, influence lipid metabolism by altering the activity of enzymes involved in lipid turnover. These alterations could potentially increase the risk of cardiovascular diseases [45]. Hormonal disruptions, including changes in insulin sensitivity, also purine metabolism, leading to hyperuricemia and further complications in kidney health and gout [46]. These complex interactions between metabolic disruptions in anemic patients may help explain the nonsignificant association observed between TG and UA levels.

The results of our study hold important implications for public health and clinical practice. The consistent association between elevated TG levels and increased UA levels, particularly among individuals aged 50–65, underscores the need for more targeted interventions aimed at managing lipid metabolism in this age group. Since both TG and UA are implicated in the development of cardiovascular diseases, insulin resistance, and gout, understanding their relationship is crucial for early identification and prevention of these conditions. Clinically, these findings suggest that monitoring and managing TG levels may provide an opportunity to reduce the risk of hyperuricemia and its related complications, especially in populations prone to metabolic disturbances, such as those aged 50–65. Furthermore, the gender-specific variations observed in the TG-UA relationship highlight the importance of considering sex as a factor when designing prevention and treatment strategies. For example, women may benefit from more personalized approaches due to their heightened sensitivity to fluctuations in lipid metabolism. Broader medical implications of this study also include the potential for using TG levels as a predictive marker for UA-related disorders, potentially allowing for more proactive management of conditions such as gout and chronic kidney disease. By incorporating these findings into clinical practice, healthcare providers can better tailor interventions to reduce the burden of metabolic and cardiovascular diseases, ultimately improving patient outcomes and public health.

## Conclusion

In this research, we employed the DDML method to investigate the effects between TG and UA. Our findings indicate a strong correlation between elevated TG levels and increased UA levels among individuals aged 50–65 years. This trend was also observed in non-anemic patients within this age group, whereas it was not significant in anemic patients. Unlike previous research on TG and UA, our research segmented hospital patients based on the condition of anemia, suggesting that strategies to reduce TG to decrease high UA should be applied primarily in non-anemic patients. In anemic patients, lowering lipid levels does not achieve the same reduction in UA levels. This distinction highlights the importance of considering anemia status when addressing hyperuricemia through TG management. Our research suggests that to address hyperuricemia, reducing TG levels may be an effective strategy. However, this strategy should not be applied to anemic patients aged 50–65.

Our research has several limitations. First, the availability of consistent and up-to-date global prevalence data on hyperuricemia was constrained, which may affect the generalizability of cross-national comparisons. Second, our heterogeneity analysis was restricted to age, sex, and blood triglyceride (TG) levels; other potential confounding factors (e.g., dietary patterns, genetic predisposition) were not systematically evaluated. Future investigations should incorporate a broader range of covariates to strengthen the robustness of such analyses.

## Author contributions

**Conceptualization:** Huaxin Zhang.

**Data curation:** Tong Zhi.

**Formal analysis:** Tong Zhi.

**Investigation:** Tong Zhi.

**Methodology:** Huaxin Zhang.

**Supervision:** Huaxin Zhang.

**Writing – original draft:** Tong Zhi.

**Writing – review & editing:** Jin Song, Shuai Zhang, Qi Li, Mingrui Li, Huaxin Zhang.

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
