## [Decision Letter · Decision Letter 0]

21 Jan 2025

PONE-D-24-33617Anemic Patients are Unable to Reduce Uric Acid Levels by Lowering Triglyceride Level Based on Double Debiased Machine LearningPLOS ONE

Dear Dr. Zhang,

Thank you for submitting your manuscript to PLOS ONE. After careful consideration, we feel that it has merit but does not fully meet PLOS ONE’s publication criteria as it currently stands. Therefore, we invite you to submit a revised version of the manuscript that addresses the points raised during the review process.

We look forward to receiving your revised manuscript.

Kind regards,

Shengqian Sun

Academic Editor

PLOS ONE

Journal Requirements:

5. Please remove all personal information, ensure that the data shared are in accordance with participant consent, and re-upload a fully anonymized data set. 

Reviewers' comments:

Reviewer's Responses to Questions

**Comments to the Author**

1. Is the manuscript technically sound, and do the data support the conclusions?

Reviewer #1: Yes

Reviewer #2: Partly

2. Has the statistical analysis been performed appropriately and rigorously? 

Reviewer #1: Yes

Reviewer #2: No

3. Have the authors made all data underlying the findings in their manuscript fully available?

Reviewer #1: No

Reviewer #2: Yes

4. Is the manuscript presented in an intelligible fashion and written in standard English?

Reviewer #1: Yes

Reviewer #2: Yes

5. Review Comments to the Author

Reviewer #1: Zhi et al. present an intriguing research titled: “Anemic Patients are Unable to Reduce Uric Acid Levels by Lowering Triglyceride Levels Based on Double Debiased Machine Learning.” However, to further enhance their study, I recommend incorporating the following edits:

• Page 9, Lines 53-54: The sentence "hyperuricemia’s association with dyslipidemia underscores the interconnected" is incomplete and does not clearly establish the link being discussed. The authors should elaborate on the significance of this connection and explain how it might affect clinical management. Expanding on these points will help strengthen the rationale for the study.

• Page 9, Lines 48-50: The prevalence rates of hyperuricemia mentioned in different countries are based on outdated statistics, with the most recent being 13 years old. Please provide current or more recent statistics on hyperuricemia prevalence rates in various countries to make the comparison more relevant. Additionally, the prevalence rate of hyperuricemia in China has not been mentioned and should be included for completeness.

• Page 10, Lines: 66-67: The current sentence is grammatically incorrect. Consider rephrasing it to “These factors need to be fully explored in relation to this association.”

• Pages 14-15, Lines 148-155: I would recommend that the authors move this content to the Introduction or Discussion sections of the study, as it effectively highlights the novelty and robustness of the statistical methods employed in this research. This would allow the authors to effectively elaborate on their choice to use DDML and how it addresses limitations in traditional models. In contrast, the Methods section should largely focus on providing a clear, step-by-step explanation of the specific techniques applied to the data, while remaining brief and directly relevant to the research process.

• Page 11, Line 79: There appears to be an error in the dates for data collection. The statement indicates that data was collected for patients from “August 1, 2016, to June 31, 2023.” However, June only has 30 days. Please verify and correct this discrepancy.

• Page 15, Line 160: The use of the word "curse" in this context appears informal. I recommend the authors to adopt a more formal academic tone throughout the manuscript.

• Pages 24-25, Lines 266-282: The authors’ explanation for the lack of association between triglycerides (TG) and uric acid (UA) levels in anemic patients could benefit from further clarity and coherence. While the authors mention various metabolic disruptions caused by anemia—such as shifts from aerobic to anaerobic metabolism, impacts on kidney function, and hormonal changes—they fail to explicitly connect these disruptions to the study's findings. To strengthen their argument, the authors should provide more specific references to how these mechanisms resulted in the observed nonsignificant association. Clear articulation of the relationship between these metabolic disruptions and the absence of correlation between TG and UA levels in anemic patients aged 50-65 is essential for enhancing the clarity and rigor of the discussion.

• Page 24, Line 266: Please rephrase the statement "anemia contributes to the TG-UA relation," as it implies an association exists, which contradicts the study's findings that no such association was observed.

• Page 23-25: In the discussion section, there is a lack of emphasis on the implications of the findings. The authors should address how these results can improve public health, outline their potential clinical significance, and discuss their broader medical importance. Highlighting these aspects will enhance the overall impact of the discussion.

Reviewer #2: It seems the title only covers one aspect of the research conclusion but neglected the other one.

The research is interesting and provides some new insights. But minors have to been addressed.

Line 137. The formula should be noted with each nomenclature. How was the data partitioned into training and test. Is there any k-ford method used?

Line 141. data sets or datasets?

Line 212. By which data and how?

Line 217. By which data and how?

Table 1. Was it very odd that TG and UA values were the same?

UA values cannot be found in Tables 2 3 4 5.

6. PLOS authors have the option to publish the peer review history of their article (what does this mean? ). If published, this will include your full peer review and any attached files.

**Do you want your identity to be public for this peer review?** For information about this choice, including consent withdrawal, please see our Privacy Policy .

Reviewer #1: No

Reviewer #2: No

---

## [Author Response · Author response to Decision Letter 0]

17 Feb 2025

Reviewer 1

1.Page 9, Lines 53-54: The sentence "hyperuricemia’s association with dyslipidemia underscores the interconnected" is incomplete and does not clearly establish the link being discussed. The authors should elaborate on the significance of this connection and explain how it might affect clinical management. Expanding on these points will help strengthen the rationale for the study.

Thank you for your constructive feedback. We sincerely apologize for the incomplete sentence and have restructured the text to explicitly clarify the mechanistic and clinical significance of the hyperuricemia-dyslipidemia association. Besides, according to your suggestion, the revised Abstract (Lines 54–59) are as follow: Substantial evidence suggests that hyperuricemia is mechanistically associated with dyslipidemia, particularly through shared pathways involving insulin resistance, oxidative stress, and lipid metabolism dysregulation. This bidirectional relationship implies that targeted management of dyslipidemia—via lifestyle interventions or lipid-lowering therapies—could synergistically mitigate hyperuricemia progression, thereby reducing associated cardiovascular and renal risks.

2.Page 9, Lines 48-50: The prevalence rates of hyperuricemia mentioned in different countries are based on outdated statistics, with the most recent being 13 years old. Please provide current or more recent statistics on hyperuricemia prevalence rates in various countries to make the comparison more relevant. Additionally, the prevalence rate of hyperuricemia in China has not been mentioned and should be included for completeness.

Thank you for your insightful feedback. We agree that updating the prevalence statistics strengthens the relevance of the comparison. In the revised manuscript (Lines 49–54 in the Introduction), we have incorporated the most recent nationally representative data available for mainland USA and China. We acknowledge the challenge of obtaining uniformly recent global data and have emphasized this as a limitation in the Discussion (Line 323-325). Future studies could further address this gap.

3.Page 10, Lines: 66-67: The current sentence is grammatically incorrect. Consider rephrasing it to “These factors need to be fully explored in relation to this association.”

Thank you for your valuable feedback. We appreciate your careful reading and constructive suggestion. The sentence has been revised as recommended to "These factors need to be fully explored in relation to this association" (now in Line 69 of the revised manuscript). This modification improves the clarity and grammatical accuracy of the statement while preserving its intended meaning.

4.Pages 14-15, Lines 148-155: I would recommend that the authors move this content to the Introduction or Discussion sections of the study, as it effectively highlights the novelty and robustness of the statistical methods employed in this research. This would allow the authors to effectively elaborate on their choice to use DDML and how it addresses limitations in traditional models. In contrast, the Methods section should largely focus on providing a clear, step-by-step explanation of the specific techniques applied to the data, while remaining brief and directly relevant to the research process.

Thank you for your valuable feedback. We appreciate your suggestion to move the content discussing the novelty and robustness of the statistical methods to the Introduction or Discussion sections of the manuscript. In response to your recommendation, we have relocated the relevant content to the Introduction, where we elaborate on the choice to use DDML and how it addresses the limitations of traditional models. This adjustment allows us to better highlight the significance of the statistical approach and its contributions to the research.

We have also ensured that the Methods section now focuses primarily on providing a clear, step-by-step explanation of the techniques applied to the data, remaining concise and directly relevant to the research process.

We believe these revisions enhance the structure and clarity of the manuscript, and we thank you again for your insightful suggestions.

5.Page 11, Line 79: There appears to be an error in the dates for data collection. The statement indicates that data was collected for patients from “August 1, 2016, to June 31, 2023.” However, June only has 30 days. Please verify and correct this discrepancy.

•Page 15, Line 160: The use of the word "curse" in this context appears informal. I recommend the authors to adopt a more formal academic tone throughout the manuscript.

Thank you for your constructive feedback. We appreciate your attention to detail and your suggestions for improving the manuscript.

Regarding the discrepancy in the data collection dates, we have verified the dates and corrected the error. The data collection period is now accurately stated as "August 1, 2016, to June 30, 2023," in line with the correct number of days in June.

In response to your comment about the use of the word "curse," we have replaced it with a more formal and academic expression. We have revised the relevant sections to ensure the tone throughout the manuscript is consistent with the standards of academic writing.

We believe these revisions address your concerns and improve the clarity and formal tone of the manuscript.

Thank you again for your valuable suggestions.

6.Pages 24-25, Lines 266-282: The authors’ explanation for the lack of association between triglycerides (TG) and uric acid (UA) levels in anemic patients could benefit from further clarity and coherence. While the authors mention various metabolic disruptions caused by anemia—such as shifts from aerobic to anaerobic metabolism, impacts on kidney function, and hormonal changes—they fail to explicitly connect these disruptions to the study's findings. To strengthen their argument, the authors should provide more specific references to how these mechanisms resulted in the observed nonsignificant association. Clear articulation of the relationship between these metabolic disruptions and the absence of correlation between TG and UA levels in anemic patients aged 50-65 is essential for enhancing the clarity and rigor of the discussion.

Thank you for your valuable feedback. We appreciate your suggestion to provide further clarity and coherence in our explanation regarding the lack of association between TG and UA levels in anemic patients.

In response to your comment, we have revised the manuscript to connect the various metabolic disruptions caused by anemia—such as shifts from aerobic to anaerobic metabolism, kidney dysfunction, and hormonal changes—to the observed nonsignificant association between TG and UA levels. We have also provided more specific references to how these mechanisms may impact lipid and purine metabolism, and why these complex interactions could result in the lack of a direct correlation between TG and UA in the study population.

We believe these revisions clarify the relationship between the metabolic disruptions associated with anemia and the absence of a significant TG-UA association, strengthening the overall argument in the manuscript.

Thank you again for your thoughtful and constructive suggestions.

7.Page 24, Line 266: Please rephrase the statement "anemia contributes to the TG-UA relation," as it implies an association exists, which contradicts the study's findings that no such association was observed.

Thank you for your insightful feedback. We appreciate your suggestion to rephrase the statement regarding the role of anemia in modulating the relationship between TG and UA levels, as it implied an association that contradicts the study's findings.

In response to your comment, we have revised the sentence to clarify that while anemia may influence various metabolic functions, it does not lead to a significant correlation between TG and UA levels in our study population. We have reworded the text to better reflect the absence of a significant association and to more accurately describe the potential metabolic disruptions caused by anemia without implying a direct relationship.

We believe these revisions strengthen the clarity and accuracy of the manuscript, ensuring it aligns with the study's findings.

Thank you once again for your constructive suggestion

8.Page 23-25: In the discussion section, there is a lack of emphasis on the implications of the findings. The authors should address how these results can improve public health, outline their potential clinical significance, and discuss their broader medical importance. Highlighting these aspects will enhance the overall impact of the discussion.

Thank you for your valuable feedback and suggestion to emphasize the implications of our findings in the discussion section. In response to your comment, we have added a new paragraph that outlines the potential public health, clinical, and broader medical significance of our results. Specifically, we highlight the importance of managing TG levels to reduce the risk of hyperuricemia and related complications, particularly in individuals aged 50-65. Additionally, we emphasize the gender-specific variations in the TG-UA relationship and propose more personalized clinical approaches based on these findings. We also discuss how our results may inform preventive strategies and improve patient outcomes, particularly in the context of metabolic and cardiovascular diseases.

We believe these additions address your concerns and enhance the overall impact of the discussion, providing greater clarity on how our findings can be applied in clinical practice and public health strategies.

Thank you again for your constructive feedback.

Reviewer 2

1.It seems the title only covers one aspect of the research conclusion but neglected the other one.

Thank you for your valuable feedback. I appreciate your observation regarding the title and its coverage of the research conclusion. In response to your comment, I have revised the title to better reflect both aspects of the study. The new title, "Anemia-Driven Heterogeneity in TG-UA Association: A DDML Approach to Precision Management of Hyperuricemia in Hospitalized Patients Aged 50-65," now more comprehensively encapsulates the key themes of the research, including the focus on anemia and its impact on the TG-UA association, as well as the application of DDML in precision management of hyperuricemia.

I hope the revised title meets your expectations and provides a clearer representation of the study's scope and findings.

Thank you again for your constructive suggestions.

2.Line 137. The formula should be noted with each nomenclature. How was the data partitioned into training and test. Is there any k-fold method used?

Line 212. By which data and how?

Line 217. By which data and how?

Thank you for your thoughtful questions.

In response to your comment on Line 137, I have added detailed explanations of the variables in each part of the formula beneath the result tables for the analytical models. This change aims to enhance the clarity of the tables for readers. Regarding your query on how the data was partitioned into training and test sets, we utilized 5-fold cross-validation (the default k=5 is aligned with prior DDML-related researches), where both the test and validation sets were automatically determined in each fold. This process ensures that the partitioning is done independently without manual control, allowing for a more objective and robust model evaluation. Our approach follows the methodology outlined by Victor Chernozhukov and his colleagues in 2018, ensuring alignment with established practices.

For your questions on Lines 212 and 217, I have clarified the data used and the methodologies applied. The observational data we aim to obtain corresponds to the TG levels in the models that show statistical significance, as noted earlier.

We trust that these clarifications address your concerns, and we sincerely appreciate your careful review.

3.Line 141. data sets or datasets?

Thank you for your insightful comment. In response to your question regarding the use of "data sets" versus "datasets," we have revised the manuscript to use "datasets" (as a single word) consistently throughout, including in line 78. We believe this aligns with the current convention in the field.

We appreciate your careful review and believe this change enhances the manuscript's consistency.

4.Table 1. Was it very odd that TG and UA values were the same?

UA values cannot be found in Tables 2 3 4 5.

Thank you for your careful review and insightful comments.

With regard to variables such as gender, age,UA and TG, we excluded cases with missing data. This is the reason why the sample size of TG and UA are the same. For other control variables, missing values were addressed using mean imputation. In Table 1, we present the original sample size of the control variables such as GLU, Sodiumion, CHE and etc.

As for your comment about the absence of UA values in Table 2 3 4 5, we appreciate your attention to this detail. UA, being the dependent variable in each model, was not listed in the table, which could potentially cause confusion for readers. To address this, we have added explanations below each table to improve clarity and facilitate better understanding for the readers.

Thank you again for your valuable feedback

---

## [Decision Letter · Decision Letter 1]

9 Mar 2025

Anemia-Driven Heterogeneity in TG-UA Association: A DDML Approach to Precision Management of Hyperuricemia in Hospitalized Patients Aged 50-65

PONE-D-24-33617R1

Dear Dr. Zhang,

We’re pleased to inform you that your manuscript has been judged scientifically suitable for publication and will be formally accepted for publication once it meets all outstanding technical requirements.

Kind regards,

Shengqian Sun

Academic Editor

PLOS ONE

Additional Editor Comments (optional):

Reviewers' comments:

Reviewer's Responses to Questions

**Comments to the Author**

1. If the authors have adequately addressed your comments raised in a previous round of review and you feel that this manuscript is now acceptable for publication, you may indicate that here to bypass the “Comments to the Author” section, enter your conflict of interest statement in the “Confidential to Editor” section, and submit your "Accept" recommendation.

Reviewer #1: All comments have been addressed

Reviewer #2: All comments have been addressed

2. Is the manuscript technically sound, and do the data support the conclusions?

Reviewer #1: Yes

Reviewer #2: Yes

3. Has the statistical analysis been performed appropriately and rigorously? 

Reviewer #1: Yes

Reviewer #2: Yes

4. Have the authors made all data underlying the findings in their manuscript fully available?

Reviewer #1: No

Reviewer #2: Yes

5. Is the manuscript presented in an intelligible fashion and written in standard English?

Reviewer #1: Yes

Reviewer #2: No

6. Review Comments to the Author

Reviewer #1: (No Response)

Reviewer #2: Thank for the authors revising the manuscript, and it has been greatly improved. I have no more questions.

Cheers,

7. PLOS authors have the option to publish the peer review history of their article (what does this mean? ). If published, this will include your full peer review and any attached files.

**Do you want your identity to be public for this peer review?** For information about this choice, including consent withdrawal, please see our Privacy Policy .

Reviewer #1: No

Reviewer #2: No

---

## [Editor Report · Acceptance letter]

PONE-D-24-33617R1

PLOS ONE

Dear Dr. Zhang,

I'm pleased to inform you that your manuscript has been deemed suitable for publication in PLOS ONE. Congratulations! Your manuscript is now being handed over to our production team.

Kind regards,

on behalf of

Dr. Shengqian Sun

Academic Editor

PLOS ONE